# Muscle Damage in Systemic Sclerosis and CXCL10: The Potential Therapeutic Role of PDE5 Inhibition

**DOI:** 10.3390/ijms22062894

**Published:** 2021-03-12

**Authors:** Clarissa Corinaldesi, Rebecca L. Ross, Giuseppina Abignano, Cristina Antinozzi, Francesco Marampon, Luigi di Luigi, Maya H. Buch, Valeria Riccieri, Andrea Lenzi, Clara Crescioli, Francesco Del Galdo

**Affiliations:** 1Leeds Institute of Rheumatic and Musculoskeletal Medicine, University of Leeds, Leeds LS9 7TF, UK; corinaldesiclarissa@gmail.com (C.C.); R.L.Ross@leeds.ac.uk (R.L.R.); G.Abignano@leeds.ac.uk (G.A.); M.Buch@leeds.ac.uk (M.H.B.); 2Department of Movement, Human and Health Sciences, University of Rome Foro Italico, 00135 Rome, Italy; cristina.antinozzi@uniroma4.it (C.A.); francesco.marampon@uniroma1.it (F.M.); luigi.diluigi@uniroma4.it (L.d.L.); 3NIHR Leeds Biomedical Research Centre, Leeds Teaching Hospitals NHS Trust, Leeds LS7 4SA, UK; 4Rheumatology Institute of Lucania (IReL), Rheumatology Department of Lucania, San Carlo Hospital of Potenza and Madonna delle Grazie Hospital of Matera, 85100 Potenza, Italy; 5Department of Radiotherapy, Sapienza University of Rome, 00185 Rome, Italy; 6Department of Internal Medicine and Medical Specialties, University Sapienza, 00185 Rome, Italy; valeria.riccieri@uniroma1.it; 7Department of Experimental Medicine, Sapienza University of Rome, 00185 Rome, Italy; andrea.lenzi@uniroma1.itc

**Keywords:** CXCL10, sildenafil, systemic sclerosis, myocytes, inflammation

## Abstract

Skeletal muscle damage is a common clinical manifestation of systemic sclerosis (SSc). C-X-C chemokine ligand 10 (CXCL10) is involved in myopathy and cardiomyopathy development and is associated with a more severe SSc prognosis. Interestingly, the phosphodiesterase type 5 inhibitor (PDE5i) sildenafil reduces CXCL10 sera levels of patients with diabetic cardiomyopathy and in cardiomyocytes. Here, we analyzed the levels of CXCL10 in the sera of 116 SSc vs. 35 healthy subjects and explored differences in 17 SSc patients on stable treatment with sildenafil. CXCL10 sera levels were three-fold higher in SSc vs. healthy controls, independent of subset and antibody positivity. Sildenafil treatment was associated with lower CXCL10 sera levels. Serum CXCL10 strongly correlated with the clinical severity of muscle involvement and with creatine kinase (CK) serum concentration, suggesting a potential involvement in muscle damage in SSc. *In vitro*, sildenafil dose-dependently reduced CXCL10 release by activated myocytes and impaired cytokine-induced Signal transducer and activator of transcription 1 (STAT1), Nuclear factor-κB (NFκB) and c-Jun N-terminal kinase (JNK) phosphorylation. This was also seen in cardiomyocytes. Sildenafil-induced CXCL10 inhibition at the systemic and human muscle cell level supports the hypothesis that PDE5i could be a potential therapeutic therapy to prevent and treat muscle damage in SSc.

## 1. Introduction

C-X-C chemokine ligand 10 CXCL10, a main interferon (IFN)-γ inducible protein, now also recognized as a type I IFN inducible biomarker, plays a critical role in the pathogenesis and progression of several autoimmune diseases, including systemic sclerosis (SSc) [1,2] systemic lupus erythematosus (SLE) [3], juvenile idiopathic arthritis (JIA) [4] and inflammatory myopathy (IM) [5]. Functionally, CXCL10 binds to C-X-C motif receptor 3 (CXCR3) and polarizes the migration of CXCR3 positive cells (i.e., T lymphocytes, natural killer cells, and monocytes) through the activation of specific intracellular signaling pathways, thereby amplifying the inflammatory events that lead to the clinical manifestations [6,7,8,9]. As recently shown, CXCL10 serum concentration is a biomarker for SSc disease progression [10]. Of note, it is reported that SSc patients with higher CXCL10 sera levels show worse skin involvement and a higher degree of clinical severity, as measured by Medsger severity scales [11,12,13]. A common feature also contributing to the worse prognosis of SSc is the involvement of muscle tissue [14,15], either limited to peripheral muscle or extended to cardiomyopathy (id est, left ventricular cardiac dysfunction and or arrhythmia), both of which have been associated with increased CXCL10 levels in the blood [16,17,18]. The theory of reducing chemokine levels as a therapeutic application for autoimmune diseases is an attractive idea [19]. We have previously reported in diabetic cardiomyopathy, a co-morbid inflammation driven condition, that CXCL10 is a pharmacologic target of sildenafil, a phosphodiesterase type 5 inhibitor (PDE5i), originally employed to treat erectile dysfunction and now more commonly used in SSc both for the management of Raynaud’s phenomenon (RP) and pulmonary artery hypertension (PAH), on the basis of its vascular effects [17,18,19,20,21].

The aim of the study herein is to explore whether CXCL10 can be a target of sildenafil at the cellular level and to inform the hypothesis of a potential increased scope of treatment with sildenafil in SSc. To this purpose, circulating CXCL10 was determined in the sera from 116 SSc patients meeting the 2013 The European League Against Rheumatism (EULAR)/American College of Rheumatology (ACR) criteria enrolled in a real life longitudinal observational study and therefore under different pharmacological treatments in combination or not with sildenafil. Patients were classified according to disease subset, auto-antibody positivity (anti-topoisomerase I (Scl70), anti-centromere and antinuclear antibodies (ACA)) and muscle involvement (skeletal muscle function: Creatin kinase, weakness, atrophy; cardiac function: Heart failure, palpitation, arrhythmia, abnormal diastolic function). Furthermore, we investigated the molecular effect of sildenafil in human myocytes secreting CXCL10 under Type 1 T helper (Th1) challenge in vitro.

## 2. Results

### 2.1. CXCL10 Serum Level Is Higher in SSc vs. Healthy Subjects and Correlates Inversely with Sildenafil Treatment 

Firstly, we measured CXCL10 blood level in the sera of healthy (*n* = 35) and SSc subjects subdivided by the inclusion or not of sildenafil within the therapeutic regimen (*n* = 17 and *n* = 99, respectively) (Figure 1A). Patients with SSc were also stratified according to disease subtype and antibody positivity (Figure 1B); characteristics of all the subjects were reported in Table 1. We showed that CXCL10 serum level was significantly lower in SSc patients taking sildenafil compared to SSc patients on no sildenafil (455.2 ± 206.1 pg/mL vs. 633.1 ± 183 pg/mL, *p* < 0.05). This latter group of SSc patients, consistent with other reports, showed higher chemokine levels vs. healthy subjects (197.5 ± 14.89 pg/mL, *p* < 0.001) (Figure 1A). Importantly, there was no significant difference in any demographic feature of patients with or without sildenafil, neither in their immune suppressive status. As expected, the proportion of patients with a history of digital ulcers was higher among patients with sildenafil since this is one of the clinical features triggering the intention to treat with sildenafil. Similarly, there were more patients with pulmonary arterial hypertension and on Bosentan among the patients taking sildenafil (Table 1). Furthermore, patients on sildenafil had a higher disease severity in most of the domains assessed. CXCL10 concentration was not different between patients with limited (LcSSc) or diffuse (DcSSc) clinical subsets or specific autoantibody (Scl70 and ACA) positivity (Figure 1B).

### 2.2. Patients with Skeletal Muscle Damage Have Higher CXCL10 Sera Levels

To determine whether CXCL10 serum concentration was associated with the presence and severity of skeletal muscle involvement, we performed sub-analysis for the SSc cohort according to the Medsger muscle severity scale and creatine kinase (CK) serum concentration. Patients were dichotomized for CK levels within or outside the normal range, characteristic of muscle damage (myositis). This showed a strong correlation between CXCL10 and CK sera levels in the patients with abnormal CK (*R*^2^ = 0.97, *p* < 0.001) (Figure 2).

### 2.3. Sildenafil Suppresses Cytokine-Induced CXCL10 Secretion in Human Skeletal Muscle Cells, with No Effect on Healthy Human Fibroblasts

Following the *ex vivo* findings, we set out to evaluate the effect of sildenafil on CXCL10 secretion by human fetal skeletal muscle cells (Hfsmc), and for comparison, by human dermal fibroblasts (Hdfbs), challenged with maximal inflammatory stimuli. Hfsmc were chosen as the muscle cell in vitro model as they have a central role in orchestrating events ending in skeletal muscle repair in myositis [22]. Hfsmc and Hdfbs were simultaneously treated with Interferon-gamma (IFNγ) (1000 U/mL) + tumor necrosis factor (TNFα) (10 ng/mL) and scalar doses of sildenafil (1 × 10^−7^, 2.5 × 10^−7^, 5 × 10^−7^, 1 × 10^−6^, 2.5 × 10^−6^, 5 × 10^−6^, 1 × 10^−5^, 2.5 × 10^−5^ M) for 24 h (Figure 3A, upper and lower left panel, respectively); drug concentrations were selected on the basis of the near therapy dose according to the pharmacokinetics (Cmax and area under the time–concentration curve, AUC). Titration experiments showed that sildenafil significantly inhibited approximately 50% of cytokine-induced CXCL10 secretion starting from 0.25 μM to 2.5 μM and at 25 μM in Hfsmc (*p* < 0.05 and *p* < 0.001, respectively, vs. IFNγ+TNFα-induced secretion, taken as 100%). Sildenafil did not show any significant impact on IFNγ+TNFα-induced CXCL10 secretion in Hdfbs. Moreover, we wanted to evaluate the effect of sildenafil (1 µM) on CXCL10 gene expression in Hfsmc and Hdfbs challenged with IFNγ (1000 U/mL) + TNFα (10 ng/mL). The CXCL10 mRNA level was significantly increased after inflammatory stimulation and sildenafil did not reduce the CXCL10 gene expression level in both Hfsmc and Hdfbs (Figure 3A, upper and lower right panel, respectively).

### 2.4. Sildenafil Inhibited STAT1, NF-kB and JNK Activation by Inflammatory Stimuli in Human Myocytes and Cardiomyocytes, Not in Human Normal Fibroblasts

To investigate whether sildenafil could target the intracellular paths underlying CXCL10 release in human skeletal muscle cells under inflammatory challenge, we evaluated STAT1, JNK and NF-kB phosphorylation induced by IFNγ+TNFα in Hfsmc, Hdfbs, and, for comparison with other striated cells, in human fetal cardiomyocytes (Hfcm) with or without sildenafil (1 μM). We found that STAT1, JNK and NF-kB phosphorylation significantly increased in all evaluated cell types stimulated with IFNγ+TNFα (Figure 3B–D). The treatment with sildenafil significantly reduced IFNγ+TNFα-induced phosphorylation in all the pathways analyzed in Hfsmc and Hfcm (*p* < 0.01 and *p* < 0.05, Figure 3B,D), while it did not change any of the cytokine-activated paths in Hdfbs (Figure 3C).

### 2.5. SSc Fibroblast CXCL10 Secretion and STAT1, JNK and NF-kB Phosphorylation Levels Are Unaffected by Sildenafil Treatment 

To evaluate the effect of sildenafil on CXCL10 protein release or mRNA expression in SSc Hdfbs, we treated SSc primary and immortalized cell lines with IFNγ (1000 U/mL) + TNFα (10 ng/mL) with/without sildenafil at scalar doses (1 × 10^−7^, 2.5 × 10^−7^, 5 × 10^−7^, 1 × 10^−6^, 2.5 × 10^−6^, 5 × 10^−6^, 1 × 10^−5^, 2.5 × 10^−5^ M) or at fixed concentration (1 × 10^−6^ M) or for 24 h (Figure 4A); drug concentrations have been selected on the basis of the near therapy dose according to the pharmacokinetics (Cmax and area under the time–concentration curve, AUC). Titration experiments showed that sildenafil did not affect CXCL10 secretion (Figure 4A left panel), nor altered CXCL10 mRNA expression in SSc Hdfbs under Th1 type challenge (Figure 4A right panel). To determine whether sildenafil could target the intracellular paths underlying CXCL10 release in SSc Hdfbs, we evaluated STAT-1, JNK and NF-kB phosphorylation with or without sildenafil (1 μM) using healthy Hdfbs for comparison. The treatment with sildenafil did not affect any of these pathways analyzed (Figure 4B). However, we observed that NF-kB and JNK phosphorylation levels were higher in SSc vs. healthy Hdfbs (albeit not significant for JNK), and STAT1 was lower in SSc, as shown by the densitometric analysis depicted beneath each blot (Figure 4B). 

### 2.6. Sildenafil Did Not Affect Viability of Human Skeletal Muscle Cells, Normal or SSc Fibroblasts

In line with previous results obtained in human cardiomyocytes, endothelial cells and immune cells, Ref. [16] sildenafil did not affect cell viability at any of tested concentrations (0.25; 1; 2.5; 25 μM) in human skeletal muscle cells, human normal or SSc fibroblasts (Figure 5A,B and C, respectively), as documented by an 3-[4,5-dimethylthiazole-2-yl]-2,5-diphenyltetrazolium bromide (MTT) test after 24 h cell exposure to the drug.

## 3. Discussion

Type I IFN inducible genes have been associated with several markers of disease activity in SSc and other autoimmune diseases. CXCL10 is one of the most studied chemokines induced by IFN type I–III [23,24,25]. Despite the “signal zero”, triggering autoimmune diseases are yet to be identified in most cases, including SSc, and the role of chemokines such as CXCL10 in amplifying and directing the inflammatory events is well characterized [26,27,28,29,30]. In the blood of SSc patients, the CXCL10 level, evaluated alone or in a composite score with CXCL11 (IFN-γ-inducible T-cell-a chemoattractant or I-TAC), is shown to be elevated in the presence of cardiac, lung, skin and muscle severe involvement [12,31]. Additionally, we have recently reported on CXCL10 serum levels as a biomarker for SSc disease progression in pre-clinical cases (cut-off value ≥ 165 pg/mL) [10]. Of note, we have previously reported on this chemokine as a systemic and local biotarget of PDE5i in diabetic cardiomyopathy [17]. The most interesting observation in the cohort herein analyzed is that SSc subjects taking sildenafil showed significantly lower CXCL10 serum levels vs. patients not assuming sildenafil and matched for other disease characteristics, including disease duration, gender, disease subset and use of immunosuppressive treatment. Further, albeit found in a small number of patients, the highest chemokine serum level was in SSc subjects presenting muscle tissue involvement, proportionally to CK serum concentration. In addition, in vitro studies document that sildenafil can specifically target human skeletal muscle cells challenged by inflammatory stimuli, significantly reducing CXCL10 release in association with the impairment of some intracellular paths activated by an inflammatory milieu. At variance with striated cells, sildenafil affected CXCL10 neither at cellular nor at intracellular level in human dermal fibroblasts from normal and SSc subjects. 

In recent years, the participation of local tissue cells, i.e., fibroblasts, cardiomyocytes and myocytes, in the pathogenesis of the inflammatory events sustaining allo- or autoimmune driven tissue inflammation and damage has been increasingly recognized [32,33]. In particular, human skeletal muscle cells under inflammatory challenge have been shown to behave as an active counterpart being a cellular source of CXCL10 [31,34,35,36]. Building on this observation, we could speculate that the high rise in CXCL10 serum level is likely due, at least in part, to myocyte secretion.

Myopathy in SSc is relatively frequent and may be an early manifestation often associated with an increased risk of myocardial disease development [37,38,39]. Clinical parameters used to define the presence of myopathy in SSc show some limitations in their predictive role; furthermore, a universally accepted gold-standard to measure therapeutic response is still lacking [39,40]. In this light and considering that CXCL10 seems involved in the early inflammatory signals in both myositis and in cardiomyopathy onset [26,27,28,29,31], we speculate that the inclusion of this chemokine within combined indexes of activity or response might be feasible and useful for evaluating/monitoring disease progression.

In addition, the previous observation depicting this chemokine as a potential bio-target of PDE5i in diabetic subjects at cardiomyopathy onset seems quite intriguing also when considering SSc. Indeed, sildenafil, a PDE5i originally licensed and used for the management of erectile dysfunction, has been successfully repurposed for the management of both refractory Raynaud’s phenomenon and as first line agent in the clinical management of PAH [20,21], extending quite widely the scope for PDE5 inhibition. More recently, PDE5i activity has been linked to other domains including cardio protection, neuroprotection and wound healing [41,42,43,44,45,46]. Many of those protective effects seem linked to its anti-inflammatory activity exerted through cGMP stabilization [47]. Accordingly, we have previously reported in vivo and in vitro on the sildenafil-induced inhibition of CXCL10 in diabetic cardiomyopathy, an inflammation-driven co-morbid condition [17].

Consistent with previous data on cardiomyocytes and endothelial cells, sildenafil inhibited CXCL10 secretion by human skeletal muscle cells exposed to proinflammatory cytokines [17]. In myocytes, this effect occurs along with a significant inhibition (about 67%) of STAT1 and JNK, NF-kB phosphorylation induced by their prototypic activators—IFNγ and TNFα, respectively. Similarly, sildenafil inhibited the same phosphorylation in cardiomyocytes, with the strongest effect exerted on NF-kB. In this context, nuclear localization studies of NF-kB and I-kB are warranted to confirm and extend our findings. 

In contrast to cardiomyocytes [17], sildenafil did not affect human skeletal muscle cell CXCL10 mRNA expression, suggesting a post-transcriptional regulatory mechanism. This is the first evidence that human skeletal muscle cells behave as cellular/intracellular targets of sildenafil in terms of CXCL10 inhibition. This observation is consistent with previous data showing reduced IFNγ and TNFα-induced CXCL10 secretion by human skeletal muscle cells when treated with immunomodulating agents, such as Infliximab, methylpred-nisolone, methotrexate, and cyclosporin A [31]. Conversely, sildenafil did not inhibit CXCL10 protein secretion and mRNA expression or any of the investigated intracellular pathways in human normal or SSc dermal fibroblasts under IFNγ+TNFα. Of interest, in line with data observed in normal Hdfbs under Th1 type challenge, we found the consistent lack of any biological response to the drug in human SSc fibroblasts primary and immortalized cells, the latter ones showing the inflammatory cascade already in activated status. This observation clearly excludes human dermal fibroblasts from sildenafil targets. So far, we speculate that sildenafil could exploit the same inhibitory function onto CXCL10 in vivo and in vitro, depending on the cell type target. Future loss of function studies (siCXCL-10) looking at the autocrine effect of CXCL10 on myoblasts or myotubes may shed further light on the role of this molecule in muscle damage.

Our study does have several limitations given the observational nature of the cohort employed and the post hoc control and analysis we have implemented. Although in patients with CK above the normal range we show a correlation between CXCL10 and CK, the sample size is very small. A larger cohort study will be needed to validate this potentially interesting observation. Furthermore, CK levels were not evaluated in the healthy controls. A further limitation of the in vitro model is the induction of CXCL10 by the use of IFNγ/TNFα, both cytokines not classically linked to SSc pathogenesis. Nevertheless, muscle inflammation during SSc is a clear inflammatory manifestation and, contrary to most other clinical manifestations, is quite sensitive to steroid treatment, not dissimilarly from polymyositis. In this sense, we could not exclude the engagement of the same known proinflammatory mechanisms in the pathogenesis of muscle damage in the context of SSc. We used validated human striatal cell systems to overcome critical limitations to in vitro studies due to the null or very limited ability of human adult striated cells to proliferate; nevertheless, further investigations must include the analysis onto skeletal muscle biopsies from SSc patients. Notwithstanding the limitations of the study, we believe this is a hypothesis generating study, suggesting the potential scope of testing, in the context of a controlled trial, the use of sildenafil to mitigate muscle damage induced by SSc. On the basis of sildenafil-induced CXCL10 inhibition at the systemic and human muscle cell level, we speculate that PDE5i might attenuate or even blunt the chemokine-dependent self-detrimental loop established between systemic circulation and local cell/tissues. This observation generates the interesting hypothesis that sildenafil could be tested in vivo as an CXCL10 inhibitor during muscle involvement, and not limited to treat digital ulcers or PAH secondary to the disease. 

## 4. Materials and Methods

### 4.1. Subjects

One hundred and sixteen consecutive patients with SSc were recruited at the scleroderma clinic within the Leeds Institute of Rheumatic and Musculoskeletal Medicine (UK). All patients signed informed consent to participate to the study and all related procedures according to the protocol REC 10/H1306-88. All patients enrolled fulfilled the 2013 EULAR/ACR classification criteria for SSc. None of the patients had overlap syndrome or were affected by any other known fibrotic condition. The following items were collected from each patient at the time of sampling: Age, gender, disease onset, disease duration at study entry, autoantibody profile, capillaroscopic pattern, presence or history of digital ulcers or pitting scars, flexion contractures, skin and internal organ involvement assessment, disease activity, disease severity, disability and current therapy. A previously published core set of clinical, laboratory and instrumental variables was applied to assess organ involvement by system [48]. Disease duration was calculated from the onset of both RP and the first non-RP symptom. Patients were classified into a limited and diffuse cutaneous subset according to LeRoy et al. [49]. Skin was assessed using the modified Rodnan skin score (mRSS) [50]. Disease activity was expressed as per the European Scleroderma Study Group activity index (EScSG-AI) [51]. The severity of organ involvement was evaluated using the Medsger’s severity scales [13]. Thirty-five healthy subjects, matched for gender and age, were also included in the CXCL10 serum analysis and analyzed for comparisons. All blood samples from subjects were collected and stored according to European League Against Rheumatism (EUSTAR) standard operating procedures [52]. CXCL10 concentration was analyzed on the Rules Based Medicine platform (Myriad, RBM, Austin, TX, USA).

### 4.2. Chemicals

Dulbecco modified eagle medium (DMEM)/Ham’s F-12 medium (1:1) with and without phenol red, phosphate buffered saline Ca2+/Mg2+-free (PBS), bovine serum albumin (BSA) fraction V, antibiotics, NaOH, EDTA—trypsin solution, Bradford reagent, PDE5 inhibitor sildenafil citrate salt were from Sigma–Aldrich Corp. (St. Louis, MO, USA). Fetal bovine serum (FBS) and fetal calf serum (FCS) were from Hyclone (Logan, UT, USA). DMEM, low glucose, GlutaMAX™ supplement, pyruvate and Lglutamine was from Thermo Fisher Scientific (Waltham, MA, USA). Recombinant human IFN-γ and recombinant human TNF-α were from Peprotech^®^ (RockyHill, NJ, USA). The ELISA kit for CXCL10 measurement was from R&D Systems (Minneapolis, MN, USA). The trypan blue 0.5% was from Euroclone^®^ (Milan, Italy). The plasticware for cell cultures and disposable filtration units for growth media preparation were purchased from Corning (Milan, Italy). The antibodies (Abs) for Western blot analysis: Rabbit polyclonal primary anti-phospho Tyr701 STAT 1 (*p*-STAT 1), mouse monoclonal primary anti-phospho Ser536 Nuclear factor-kB (*p*-NF-kB), rabbit polyclonal primary anti-STAT 1 were from Cell Signaling (Danvers, MA, USA); rabbit polyclonal primary anti-phospho Thr183/Tyr185 JNK (*p*-JNK), rabbit polyclonal primary anti-JNK/SAPK1, peroxidase secondary Abs, all reagents for SDS-PAGE were from Millipore (Billerica, MA, USA); mouse monoclonal primary anti-β actin, rabbit polyclonal anti-human primary anti-NF-kB p65 (C-20) were from Santa Cruz Biotechnology (Santa Cruz, CA, USA).

### 4.3. Cell Cultures

Human fetal skeletal muscle cells (Hfsmc) and human fetal cardiomyocytes (Hfcm) were isolated and processed as previously reported [6,31]. The use of human fetal tissue for research purposes conforms with the principles outlined in the Declaration of Helsinki and was approved by the Committee for investigation in humans of the Azienda Ospedaliero-Universitaria Careggi, Florence, Italy (protocol *n*° 6783-04, released 01/02/2002). All samples have been handled in the same way and maintained in ice-cold PBS until processed for culture preparation [6,31]. Confluent cell cultures were split into a 1:2–1:4 ratio using EDTA–trypsin solution (0.2–0.5%) and used from 3rd/8th passage. Human dermal fibroblasts (Hdfbs) were isolated from excisional skin biopsies from 7 patients with early diffuse cutaneous SSc (dcSSc) and 5 healthy controls at the SSc clinic within the Leeds Institute of Rheumatic and Musculoskeletal Medicine (UK). Full informed consent was obtained from all the subjects and approved by National Research Ethics Service (NRES) Committee (REC 10/H1306/88) and processed as previously described [53]. Briefly, the skin biopsy samples were minced with a scalpel, enzymatically disassociated with 1 mg/mL of trypsin at 37 °C in a humidified atmosphere of 5% CO_2_ for 2 h and centrifuged at 220× *g* for 10 min. After centrifugation, the supernatant was discarded, the tissue was placed in plastic culture dishes and covered with DMEM medium supplemented with 20% Fetal Calf Serum (FCS), 100 units/mL of penicillin, 100 ng/mL of streptomycin at 37 °C in a humidified atmosphere of 5% CO_2_. Fresh growth medium was added after 4 days and subsequently replenished on day 6 when a visible outgrowth of cells was obtained. Fibroblasts were used at passage 3–5. SSc fibroblasts were also immortalized with the human telomerase reverse transcriptase (hTERT) to extend their lifespan, as previously described in normal or transformed cells [54,55]. Cell cultures in their specific growth medium were maintained in a fully humidified atmosphere of 95% air and 5% CO_2_.

### 4.4. Cytokine Secretion Assay

For CXCL10 secretion assay in Hfsmc, 4000 cells/well were seeded onto 96-well flat bottom plates and maintained for 24 h in DMEM/Ham’s F-12 medium (1:1) supplemented with 10 % Fetal Bovine Serum (FBS), 100 units/mL of penicillin, and 100 ng/mL of streptomycin. After overnight starvation (medium without serum and without phenol red), cells were stimulated for 24 h with IFNγ (1000 U/mL) and TNFα (10 ng/mL) with or without sildenafil (1 μM), in serum free medium with 0.1% BSA. Cells in serum-free medium containing 0.1 % BSA and the vehicle were used as the control. The drug concentration was selected based on its near-therapeutic doses, according to its pharmacokinetics (Cmax and area under the curve, AUC). For dose–response assays, after overnight starvation, Hfsmc were incubated for 24 h with IFNγ (1000 U/mL) + TNFα (10 ng/mL) with or without different concentrations of sildenafil (1 × 10^−7^, 2.5 × 10^−7^, 5 × 10^−7^, 1 × 10^−6^, 2.5 × 10^−6^, 5 × 10^−6^, 1 × 10^−5^, 2.5 × 10^−5^ M) as previously published [17]. Supernatants were harvested, centrifuged, and stored to −20 °C until analyzed. All experiments were performed in triplicate with at least three different cells preparations.

### 4.5. Elisa Assays

CXCL10 levels were measured in cell culture supernatants using commercially available kits (R&D Systems), according to the manufacturer’s recommendations. The sensitivity ranged from 0.41 to 4.46 pg/mL for CXCL10. The intra- and inter-assay coefficients of 49 variation were 3.1 and 6.7% for CXCL10. Quality control pools of low, normal, and high concentrations for all parameters were included in each assay. The amount of secreted cytokines was normalized with protein measurement using Bradford Reagent.

### 4.6. Western Blot Analysis

For protein analysis, 1 × 10^6^ Hfsmc, Hfcm, normal dermal fibroblasts and dermal fibroblasts from patients with SSc were seeded onto 10-cm dishes and maintained for 24 h in 10 mL DMEM/Ham’s F-12 medium (1:1) with 10% FBS. After overnight starvation (medium without serum and without phenol red), the cells were stimulated for 10 min with IFNγ (1000 U/mL) and TNFα (10 ng/mL) with or without sildenafil (1 μM), in serum free medium with 0.1% BSA. Protein concentration measurement was performed with Bradford Reagent. Protein aliquots (20 μg) were processed, loaded onto 10% SDS-PAGE, transferred on nitrocellulose membranes, and incubated with primary Abs appropriately diluted in tris buffered saline—0.1% Tween 20 (TBST)—followed by Horseradish Peroxidase (HRP) secondary Abs. Proteins were revealed by the enhanced chemiluminescence system (ECL plus; Millipore). Image acquisition was performed with Image Quant Las 4000 software (GE Healthcare, Buckinghamshire, UK). Western blot analysis was performed for three independent experiments with different cell preparation.

### 4.7. RNA Extraction, Reverse Transcription and Real-Time Quantitative PCR

50,000 Hfsmc, healthy Hfbs and SSc Hfbs were seeded and maintained in the same conditions as previously reported [56,57]; after 12 h starvation (medium without serum and without phenol red), cells were stimulated for 24 h with a combination of IFNγ (1000 U/mL) and TNFα (10 ng/mL) with or without sildenafil (1 μM), in serum-free medium with 0.1% BSA, cells in serum-free medium containing 0.1% BSA and the vehicle were used as the control. Total RNA was extracted from cultured cells using TRIzol^®^ RNA Isolation Reagents (Invitrogen, Waltham, MA, USA) according to the manufacturer’s instructions. Single-stranded cDNA was obtained by reverse transcription of 1 μg of total RNA. RT-qPCRs were performed using 7500 Real Time System (Applied Biosystems^®^, Waltham, MA, USA) with SYBR-green fluorophore. Fluorescence intensities were analyzed using the manufacturer’s software (7500 Software v.2.05) and relative amounts were obtained using the 2−∆∆Ct method and normalized for the ß-actin. Data are expressed as fold increase vs. IFNγ+TNFα- induced expression taken as 1. Primers for CXCL10 were: Forward (TTCCTGCAAGCCAATTTTGT) and reverse (ATGGCCTTCGATTCTGGATT); for β-actin, forward (CTGAACCCCAAGGCCAAC) and reverse (AGCCTGGATAGCAACGTACA).

### 4.8. Cell Viability

For cell viability assays, 4000 Hfsmc, healthy Hfbs and SSc Hfbs were seeded in 96-well plates, maintained in phenol red- and serum-free medium overnight and incubated in serum-free medium containing 0.1% BSA with sildenafil (0.25, 1, 2.5 and 25 µM) for 24 h; cells in serum-free medium containing 0.1% BSA and vehicle were used as the control. Cell viability was assessed by MTT (3-(4,5-dimethylthazolk-2-yl)-2,5-diphenyl tetrazolium bromide). The MTT (5 mg/mL) was added to each well and incubated for 2 h at 37 °C. The formazan product was dissolved by adding 200 μL of DMSO to each well. The MTT absorbance value was detected at 560 nm with a micro plate reader (Bio-Rad, Hercules, CA, USA). Experiments were performed in triplicate with different cell preparations.

### 4.9. Statistical Analysis

The statistical analysis was performed using GraphPad Prism 7 software (GraphPad 50 Software, Inc., La Jolla, CA, USA). Pearson’s correlation was used to analyze the association between all studied parameters. Continuous data were compared using an unpaired Student’s *t* test or Mann–Whitney test when appropriate. Binomial data were compared using chi-square analysis and Fisher’s exact test when appropriate. Data were expressed as the mean ± standard error (SE). A *p* value less than 0.05 was considered significant and was corrected for comparisons using the Dunnett’s or Bonferroni’s post hoc test where appropriate.

## Figures and Tables

**Figure 1 ijms-22-02894-f001:**
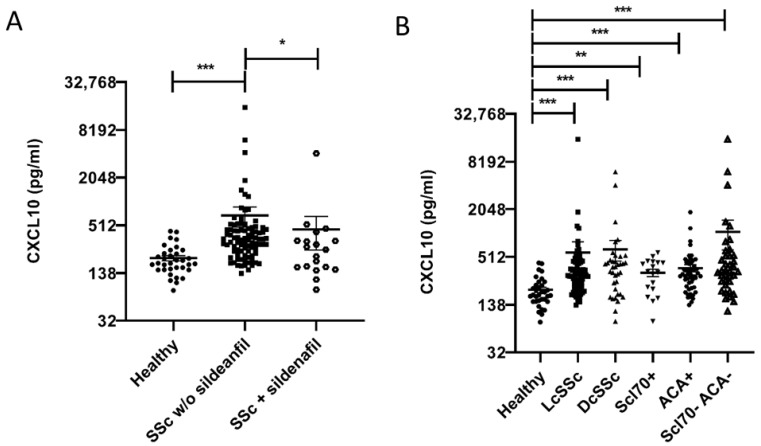
Circulating level of CXCL10 is higher in systemic sclerosis (SSc) patients than healthy subjects. Treatment with sildenafil significantly reduced CXCL10 serum level in SSc subjects. (**A**) CXCL10 serum level was significantly elevated in SSc patients without sildenafil (*n* = 99) in their therapeutic regimen vs. sildenafil treated-SSc patients (*n* = 17) and healthy subjects (*n* = 35) (633.07 ± 183.02 pg/mL vs. 455.21 ± 211.8 pg/mL and 197.5 ± 14.89 pg/mL, respectively, * *p* < 0.05 and *** *p* < 0.001). (**B**) CXCL10 was analyzed in the sera of SSc patients divided in different subgroups: Limited (LcSSc) (*n* = 72) (575.21 ± 8.49 pg/mL), diffuse (DcSSc) (*n* = 37) (640.73 ± 186.33 pg/mL), anti-topoisomerase I (ScL70) positive (*n* = 18) (324.22 ± 34.10 pg/mL), anti-centromere (ACA) positive (*n* = 51) (370.24 ± 39.81 pg/mL) and ScL70-ACA double negative (*n* = 38) (1062.74 ± 435.36 pg/mL). All the subgroups analyzed showed CXCL10 serum levels significantly higher than healthy subjects (** *p* < 0.01 and *** *p* < 0.001, respectively). No significant differences within SSc subgroups were noted. Data are expressed as CXCL10 serum level, in pg/mL (mean ± standard error of the mean (SE)).

**Figure 2 ijms-22-02894-f002:**
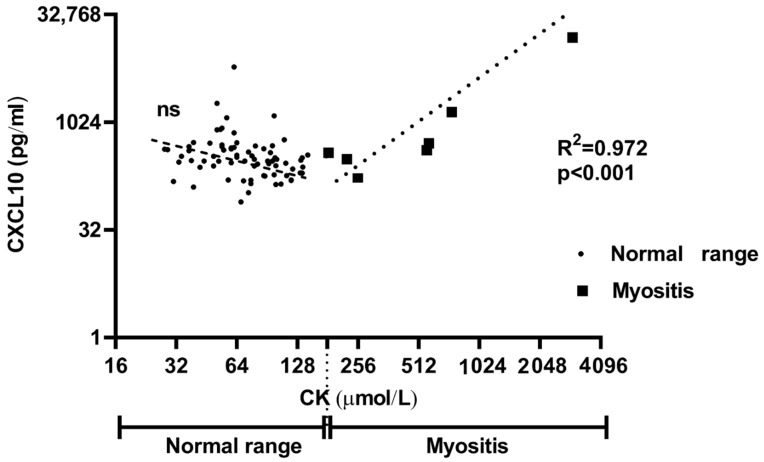
CXCL10 serum level positively correlates with an above normal range of creatine kinase (CK) blood level in SSc patients. The scatterplot has the CK blood level in μmol/L on the horizontal (X) axis, and the CXCL10 serum level, in pg/mL, on the vertical (Y) axis. CK < 180μ mol/L is considered the cut-off of normal range. Each individual is identified by a single point (dot) on 180 mol/L; each square symbolizes SSc–myositis patient with CK > 180 μmol/L (above normal range). SSc patients with a CK > 180 μmol/L showed higher levels of circulating CXCL10. The relationship between CK blood level and serum CXCL10 levels in SSc–myositis patients is strong, linear, and positive with a Pearson’s correlation coefficient of +0.97 (*p* < 0.001). No significant correlation is observed in SSc patients with CK within normal range (*R*^2^ = 0.02, *p* = 0.245).

**Figure 3 ijms-22-02894-f003:**
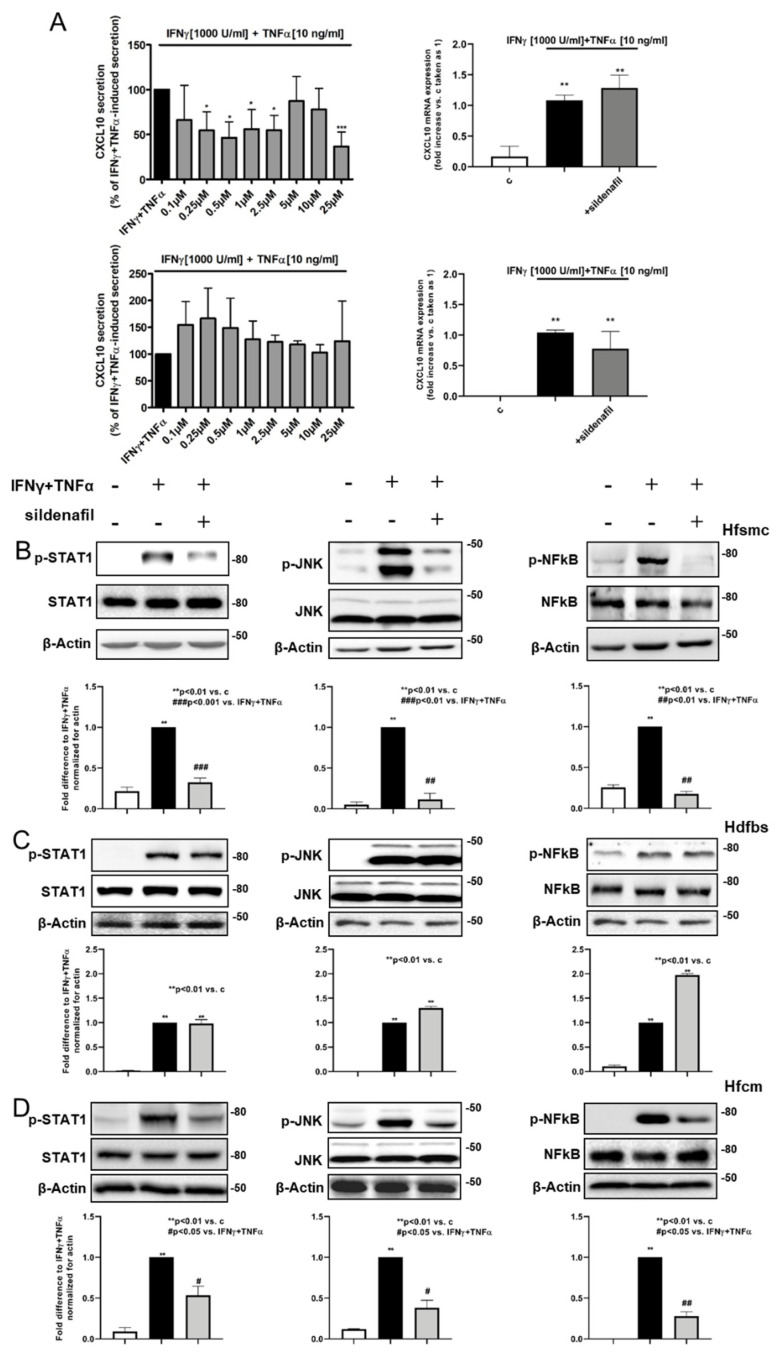
Cytokine-induced CXCL10 release in muscle cells is significantly diminished by sildenafil. Effect of sildenafil STAT 1, JNK and NF-kB phosphorylation in human myocytes (Hfsmc), dermal fibroblasts (Hdfbs) and cardiomyocytes (Hfcm). (**A**) Incubation with increasing doses of sildenafil (1 × 10^−7^, 2.5 × 10^−7^, 5 × 10^−7^, 1 × 10^−6^, 2.5 × 10^−6^, 5 × 10^−6^, 1 × 10^−5^, 2.5 × 10^−5^ M) for 24 h significantly albeit not strictly dose-dependently reduced CXCL10 protein release by Hfsmc (upper left panel) stimulated with interferon (IFN)γ (1000 U/mL) + tumor necrosis factor (TNFα) (10 ng/mL) starting from 2.5 × 10^−7^ M of sildenafil dose (54.61 ± 19.05 % of CXCL10 secretion * *p* < 0.05). Maximal inhibitory effect was at 2.5 × 10^−5^ M of sildenafil concentration (36.55 ± 6.61 % of CXCL10 secretion, ** *p* < 0.01). Data are obtained from three different experiments using different cell preparations. 24 h treatment with IFNγ (1000 U/mL) + TNFα (10 ng/mL) significantly upregulated CXCL10 mRNA expression in Hfsmc and Hdfbs; sildenafil (1 μM) did not exert any significantly change on CXCL10-cytokines induced gene level in both cell types (upper and lower right panels). CXCL10 mRNA expression is determined by RT-qPCR and is expressed as fold increase vs. IFNγ+TNFα-induced expression taken as 1 (mean ± SE). Data are obtained from three different experiments using three different cell preparations. (**B**) Western blot analysis in Hfsmc demonstrated that IFNγ+TNFα significantly increased STAT 1, JNK and NF-kB phosphorylation (** *p* < 0.01 vs. c) and the treatment with sildenafil significantly reduced all of these phosphorylation (### *p* < 0.001 and ## *p* < 0.01 vs. IFNγ+TNFα, respectively). (**C**) Western blot analysis in human healthy dermal fibroblasts revealed that IFNγ+TNFα significantly increased STAT 1, JNK and NF-kB phosphorylation (** *p* < 0.01 and * *p* < 0.05 vs. c, respectively) but the treatment with sildenafil did not affect pathway activation. (**D**) Western blot analysis in human fetal cardiomyocytes (Hfcm) confirmed the results obtained in muscle cells and showed that IFNγ+TNFα significantly increased STAT 1, JNK and NF-kB phosphorylation (** *p* < 0.01 and * *p* < 0.05 vs. c, respectively) and the treatment with sildenafil significantly reduced all these path phosphorylation (# *p* < 0.01 and ## *p* < 0.01 vs. IFNγ+TNFα, respectively). Figure 3, panel B, C and D: The densitometric analysis of band intensities is reported under each panel. Results are derived from three to five experiments, using distinct cell preparations. Data are expressed as ratio phosphorylated/β-actin fold increase vs. IFNγ+TNFα, taken as 1 (mean ± standard error of the mean (SE)).

**Figure 4 ijms-22-02894-f004:**
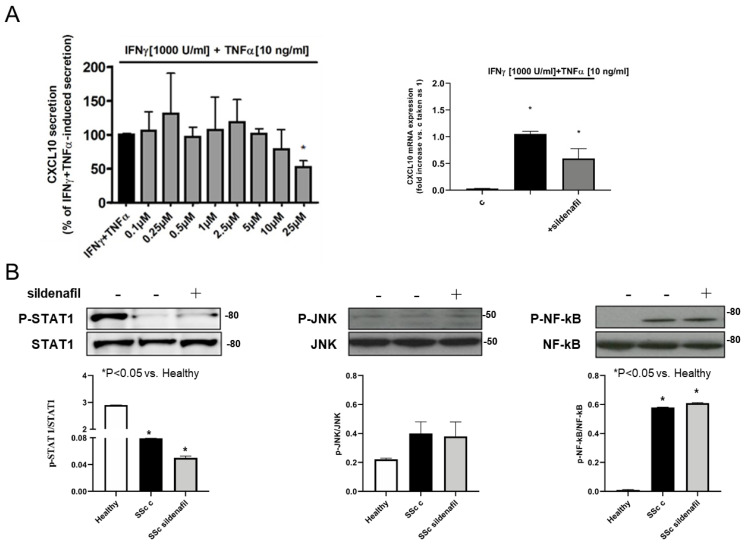
Cytokine-induced CXCL10 secretion, mRNA expression and STAT1, JNK and NF-kB phosphorylation do not change in human SSc fibroblasts after sildenafil. (**A**) Secretion and gene expression were evaluated in SSc human dermal fibroblasts challenged with IFNγ (1000 U/mL) + TNFα (10 ng/mL) and with/without sildenafil scalar doses (1 × 10^−7^, 2.5 × 10^−7^, 5 × 10^−7^, 1 × 10^−6^, 2.5 × 10^−6^, 5 × 10^−6^, 1 × 10^−5^, 2.5 × 10^−5^ M) or fixed dose (1 × 10^−6^ M), respectively, for 24 h. Sildenafil neither essentially reduced CXCL10 secretion in SSc fibroblasts (left panel) nor reduced CXCL10 mRNA expression, significantly upregulated after inflammatory stimuli (right panel). (**B**) Western blot analysis documented that sildenafil did not affect the basal level of STAT1, JNK and NF-kB phosphorylation in SSc human dermal fibroblasts. At variance with STAT1, both JNK and NF-kB basal phosphorylation showed the trend to be higher in SSc vs. healthy cells (* *p* < 0.05). Densitometric analysis of band intensities is depicted as relative density (arbitrary units) under each panel and normalized against optical density of STAT1, NF-kB and JNK total protein, respectively. Results are derived from three different experiments, using distinct cell preparations.

**Figure 5 ijms-22-02894-f005:**
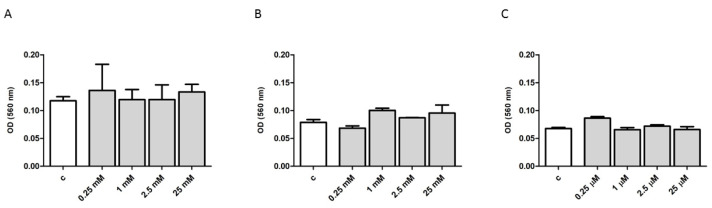
Sildenafil does not exert any effect onto the cell viability of human skeletal muscle cells (Hfsmc), normal and SSc fibroblasts. A, B and C depict cell viability in Hfsmc, healthy and SSc human dermal fibroblasts, respectively, as evaluated with an MTT test after 24 h treatment with 0.25; 1; 2.5; 25 μM of sildenafil. Results are reported as change in absorbance at 560 nm. Results are derived from three different experiments, using distinct cell preparations.

**Table 1 ijms-22-02894-t001:** Clinical features of the 116 SSc patients. Data are shown as mean ± standard deviation (SD), as median (range) or as number (percentage). The number in parentheses refers to the percentage of patients with the specific feature among the total patients with the available test results. Difference between groups were analyzed by Fisher’s exact test and unpaired *t*-test or Mann–Whitney test as appropriate. Clinically relevant criteria highlighted in bold.

Characteristic	All *n* = 116	Sildenafil *n* = 17	No Sildenafil *n* = 99	*p*
Male gender	18 (15.5)	4 (23.5)	14 (15.5)	ns
Age, years	57.2 (12.8)	52.8 (11.9)	57.9 (12.8)	ns
Disease duration from RP, years	17.2 (14.6)	20.1 (14.8)	16.7 (14.6)	ns
Disease duration from 1st non RP, years	10.2 (8.2)	12.9 (6.8)	9.8 (8.4)	ns
Disease subset (DcSSc)	38 (32.8)	8 (47.1)	30 (30.3)	ns
ANA +	113 (97.4)	16 (94.1)	97 (98)	ns
ACA +	53 (45.7)	8 (47.1)	45 (45.5)	ns
Anti-topoisomerase I +	18 (15.5)	4 (23.5)	14 (14.1)	ns
mRss	2 (0–34)	4 (0–16)	2 (0–34)	ns
Raynaud’s Phenomenon	110 (94.8)	17 (100)	93	ns
**Digital ulcers**	**18 (15.5)**	**8 (47.1)**	**10 (10.2)**	**0.0008**
Telangectasias	90 (77.6)	15 (88.2)	75 (77.3)	ns
Synovitis	9 (7.8)	1 (5.9)	8 (8.2)	ns
**Flexion contractures**	**52 (44.8)**	**13 (76.5)**	**39 (39.4)**	**0.0071**
Tendon friction rubs	5 (4.3)	1 (5.9)	4 (4)	ns
Proximal muscle weakness	8 (6.9)	0 (0)	8 (8.1)	ns
Serum CK elevation	5 (4.3)	0 (0)	5 (5)	ns
Reflux/dysphagia	89 (76.7)	12 (70.6)	77 (77.8)	ns
Early satiety/vomiting	31 (26.7)	6 (35.3)	25 (25)	ns
Diarrhoea/constipation/bloating	32 (27.6)	8 (47.1)	24 (24.2)	ns
Dyspnoea	69 (59.5)	11 (64.7)	56 (56.6)	ns
Chest HRCT fibrosis	36 (50)	7 (58.3)	29 (48.3)	ns
Restrictive defect (FVC, DLCO)	29(25.9)	6 (42.9)	23 (24)	ns
**Confirmed PAH (by RHC)**	**8 (6.9)**	**4 (23.5)**	**4 (4)**	**0.0157**
Palpitations	27 (23.3)	7 (41.2)	20 (20.2)	ns
Conduction defects	1 (1)	0 (0)	1 (1.1)	ns
SV arrhythmias	2 (1.9)	1 (7.7)	1 (1.1)	ns
V arrhythmias	1 (1)	0 (0)	1 (1.1)	ns
Diastolic dysfunction	59 (54.6)	7 (41.2)	52 (57.1)	ns
Reduced ejection fraction	8 (7.1)	1 (9.1)	7 (7.4)	ns
Arterial hypertension	17 (14.7)	3 (17.7)	14 (14.3)	ns
Renal crisis	1 (0.9)	0 (0)	1 (1)	ns
**SSc capillary pattern**	**56 (90.3)**	**11 (100)**	**28 (54.9)**	**0.0046**
- Early	13 (20.9)	1 (9.1)	12 (23.5)	ns
- Active	21 (33.9)	4 (36.4)	17 (33.3)	ns
- Late	22 (35.5)	6 (54.6)	16 (31.4)	ns
EScSG-AI	1.5 (0–7.5)	1.5 (0–4)	1 (0–7.5)	ns
Sev_general	0 (0–3)	0 (0–2)	0 (0–3)	ns
**Sev_peripheral vascular**	**1 (1–3)**	**2 (0–3)**	**1 (0–3)**	**0.0059**
Sev_skin	1 (0–3)	1 (0–2)	1 (0–3)	ns
Sev_joint/tendon	0 (0–4)	1 (0–2)	0 (0–4)	0.03
**Sev_muscle**	**0 (0–3)**	**1 (0–2)**	**0 (0–3)**	**<0.0001**
Sev_GI tract	1 (0–2)	1 (0–2)	1 (0–2)	ns
**Sev_lung**	**2 (0–4)**	**1 (0–2)**	**2 (0–4)**	**0.0003**
**Sev_heart**	**0 (0–3)**	**1 (0–2)**	**0 (0–3)**	**<0.0001**
**Sev_kidney**	**0 (0–1)**	**1 (0–2)**	**0 (0–1)**	**<0.0001**
**Sev_total**	**6 (2–19)**	**9 (2–12)**	**6 (2–19)**	**0.047**
Immunosuppressive/anti-rheumatic therapy	46 (39.7)	5 (29.4)	41 (35.3)	ns
**Sildenafil**	**17 (14.7)**	**17**	**0**	**<0.0001**
**Bosentan**	**4**	**3**	**1**	**0.0097**
Prostanoids	30	7	23	ns

ANA, antinuclear antibodies; ACA, anti-centromere antibodies; CK, creatine kinase; DcSSc, diffuse cutaneous systemic sclerosis; DLCO, diffusion lung capacity of carbon monoxide; EScSG-AI, European Scleroderma Study Group–activity index; FVC, forced vital capacity; GI, gastrointestinal; HRCT, high resolution computed tomography; mRSS, modified Rodnan skin score; PAH, pulmonary arterial hypertension; RP, Raynaud’s phenomenon; Sev, Severity; SSc, systemic sclerosis. Clinical features with a *p*-value < 0.05 are showed in bold; ns, not significant.

## Data Availability

All data shared within manuscript.

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
