# Peer review of "Muscle Damage in Systemic Sclerosis and CXCL10: The Potential Therapeutic Role of PDE5 Inhibition"

_ijms, 2021, doi:10.3390/ijms22062894_

Round 1

Reviewer 1 Report

In this unbalanced observational study, Corinaldelsi et al. aimed to correlate levels of chemokine CXCL 10 with markers of muscle damage in patients who present systemic sclerosis (SSc) and investigate whether patients with SSs benefit from treatment with the phosphodiesterase type 5 inhibitor sildenafil treatment. In another set of in vitro experiments, the group attempts to identify the underlying mechanisms of the sildenafil mediated reduction in CXCL 10 in three cell types.

The present reviewer sees a few significant issues in both in vivo and in vitro models:

In vivo model:

The patients on sildenafil fared clinically worse than the other patients on a different medication. Therefore, the authors could not test the hypothesis that PDE5i in the form of sildenafil could be a potential therapeutic intervention to prevent and treat muscle damage in SSc.

Furthermore, the authors state that they explored the effect of CVCL10 serum levels in 116 SSc patients and 35 healthy patients. This statement is rather bold, as only 17 SSc of the total 116 patient cohort received the drug while the control subjects did not receive any of the drugs.

There is no in-depth investigation, such as details of histochemistry evidence, to document the presence/changes in muscle damage. The authors rely entirely on the circulating CPK enzyme levels. Figure 2 is crucial to the authors' contention that chemokine CXCL 10 levels correlate with muscle damage markers in patients who present systemic sclerosis (SSc). A closer inspection of the data in Fig. 2A shows that the strong correlation, but not necessarily implying causality, was achieved due to the data's skewed nature. Most of the data were tightly concentrated around a mid-point, which, together with an extreme value (outlier?), generates a strong positive correlation. In fact, Figure 2B support the reviewer's claim as if four extreme values from the myositis group are removed, then the correlation between CXCL10 and CPK become negative, maybe even significant.

Most of the content in Table 1 is irrelevant to support any conclusion. The authors should limit the presentation only to the data significantly different between the patients with and without the drug.

How did the CPK levels in patients relate to the CKP in the healthy controls?

In vitro model:

How representative are the results reported from the undefined-muscle myocytes compared to the closer to muscle definition, but presently missing, myotubes?

Figs. 3A (top) and B (bottom) graphs display the curve of the effects of sidenafil on CXCL10 in muscle myocytes and fibroblasts. The range (max-min) variation in both distributions looks similar (~50%). Yet, the significant levels are attained intriguingly only for the myocytes. 

The effect of the sildenafil titration curve on CXCL10 response levels in cardiomyocytes is missing. Yet, we are provided with unnecessary curve dose responses in human fibroblasts. What was the judgement behind this choice? Overall, the reporting of the experiments in fibroblasts could be altogether omitted under this form. It would be satisfactory if the authors mentioned the fibroblast results in the text as negative results.

Interestingly, the authors were able to identify at gene and protein levels an isoform of actin, ß-actin, which is typically not expressed in muscle. Do the authors have any external evidence for these findings? On a different note, it became apparent that the authors did not report a molecular protein marker in the WB results.

The font of the labels on several y-axes is at minuscule levels. It is impossible to identify the numeric value of the y-variables.

Author Response

In vivo model:

  1. The patients on sildenafil fared clinically worse than the other patients on a different medication. Therefore, the authors could not test the hypothesis that PDE5i in the form of sildenafil could be a potential therapeutic intervention to prevent and treat muscle damage in SSc.

This is absolutely correct, we apologise if the manuscript as written gave a different impression. This is not a clinical trial for PDE5i in SSc muscle damage. Here we aimed simply to describe an observational cohort and report serum concentration and CK and chemokines in patients with different intervention, to generate an hypothesis rather than to test it. We removed any mention that could give a different impression to the reader. All changes can be seen in the track-change version of the manuscript.

  1. Furthermore, the authors state that they explored the effect of CVCL10 serum levels in 116 SSc patients and 35 healthy patients. This statement is rather bold, as only 17 SSc of the total 116 patient cohort received the drug while the control subjects did not receive any of the drugs.

We agree the statement may generate confusion and we reworded accordingly :

Abstract: Here we analysed the levels of CXCL10 in the sera of 116 SSc vs. 35 healthy subjects, and explored differences in 17 SSc patients on stable treatment with Sildenafil

  1. There is no in-depth investigation, such as details of histochemistry evidence, to document the presence/changes in muscle damage. The authors rely entirely on the circulating CPK enzyme levels. Figure 2 is crucial to the authors' contention that chemokine CXCL 10 levels correlate with muscle damage markers in patients who present systemic sclerosis (SSc). A closer inspection of the data in Fig. 2A shows that the strong correlation, but not necessarily implying causality, was achieved due to the data's skewed nature. Most of the data were tightly concentrated around a mid-point, which, together with an extreme value (outlier?), generates a strong positive correlation. In fact, Figure 2B support the reviewer's claim as if four extreme values from the myositis group are removed, then the correlation between CXCL10 and CPK become negative, maybe even significant.

As the reviewer is surely aware, increase in CK does not trigger necessarily a muscle biopsy in patients with Scleroderma as it is considered a disease manifestation with no need for further diagnosis in most cases. To avoid confusion we replaced “muscle damage” through the manuscript with “ increased CK serum level, which can be read as a sign of muscle damage”.

As far as the nature of the correlation we observed the skewing indeed. Since these are consecutive patients removing any of the data would be data manipulation, against scientific rigor, hence we simply described the data as they are. Nevertheless, to acknowledge this skewing and invite interested reader to replicate our findings we included the following paragraph in the discussion:

“Although in patients with CK above normal range we show correlation between CXCL10 and CK. The sample size is very small. A larger cohort study will be needed to validate this potentially interesting observation.”

  1. Most of the content in Table 1 is irrelevant to support any conclusion. The authors should limit the presentation only to the data significantly different between the patients with and without the drug.

We thank the reviewer for this comment. Nevertheless, the clinical reader may wonder what were other differences of the groups that could potentially be associated with the differences in CXCL-10 and in this view the negative data are as important as the positive one. To address the reviewer’ concern and focus the attention of the reader, we highlighted in bold the significant differences.

  1. How did the CPK levels in patients relate to the CKP in the healthy controls?

This is a very important point raised by the reviewer. Healthy controls were not assessed for CK as these were healthy controls of the chemokine study and not of the overall observational cohort. We acknowledge this limitation in the discussion of the revised manuscript:

'Furthermore, CK levels were not evaluated in the healthy controls.'

We also updated in methods.

In vitro model:

  1. How representative are the results reported from the undefined-muscle myocytes compared to the closer to muscle definition, but presently missing, myotubes?

We understand the concern about using myoblasts instead of myotubes, which are closer to muscle definition. Nevertheless, phenotypically mature myoblasts are reported to be the cell population with a central role in orchestrating events ending in skeletal muscle repair in myositis (J. Neuroimmunol. 200, 62–70, doi.org/10.1016/j.jneuroim.2008.06.012). Hfsmc express motor and structural proteins typical of mature skeletal muscle cell phenotype (with the ability to spontaneously fuse in myotubes) thus representing an optimal human cell model to study muscular diseases, from myositis to metabolic disease (often overlapping) as shown in previous original papers (Eur J Cell Biol. 2012 Feb;91(2):139-49. doi: 10.1016/j.ejcb.2011.09.011; J Endocrinol Invest. 2013 Dec;36(11):1020-6. doi: 10.3275/9034; PLoS One. 2013 Oct 30;8(10):e77745. doi: 10.1371/journal.pone.0077745; J Steroid Biochem Mol Biol. 2017 Mar;167:169-181. doi: 10.1016/j.jsbmb.2016.12.010; J Endocrinol Invest. 2017 Oct;40(10):1133-1143. doi: 10.1007/s40618-017-0686-y; Endocrine. 2018 Mar;59(3):602-613. doi: 10.1007/s12020-017-1378-2; J Endocrinol Invest. 2019 Aug;42(8):897-907. doi: 10.1007/s40618-018-0998-6).

This has been advised in the revised manuscript within the results, see page 6.

"Hfsmc were chosen as the muscle cell in vitro model as they have a central role in orchestrating events ending in skeletal muscle repair in myositis (J. Neuroimmunol. 200, 62–70, doi.org/10.1016/j.jneuroim.2008.06.012).."

  1. Figs. 3A (top) and B (bottom) graphs display the curve of the effects of sildenafil on CXCL10 in muscle myocytes and fibroblasts. The range (max-min) variation in both distributions looks similar (~50%). Yet, the significant levels are attained intriguingly only for the myocytes. 

T tests were performed on the triplicate studies (each dose of sildenafil, and not the dose response) and only statistical significance achieved in myocytes and not fibroblasts. CXCL10 secretion from fibroblasts is not reduced by sildenafil treatment as the data never goes below 100% of IFNgamma/TNFalpha treatment. Whereas for myocytes, there is a clear reduction, albeit not in all doses of sildenafil.

  1. The effect of the sildenafil titration curve on CXCL10 response levels in cardiomyocytes is missing. Yet, we are provided with unnecessary curve dose responses in human fibroblasts. What was the judgement behind this choice? Overall, the reporting of the experiments in fibroblasts could be altogether omitted under this form. It would be satisfactory if the authors mentioned the fibroblast results in the text as negative results.

We have previously shown that cardiomyocytes and endothelial cells are cellular targets of sildenafil in terms of CXCL10 release inhibition (Di Luigi L., Inflammation. 39(3):1238-52. doi: 10.1007/s10753-016-0359-6), thus have not included this work here. There is much focus on dermal fibroblasts as the key driver of SSc phenotype, including muscle involvement, thus we provided corresponding analysis.

On page 9 of discussion we have included the above reference with our comment that: Consistent with previous data on cardiomyocytes and endothelial cells, sildenafil inhibited CXCL10 secretion by human skeletal muscle cells exposed to proinflammatory cytokines.

  1. Interestingly, the authors were able to identify at gene and protein levels an isoform of actin, ß-actin, which is typically not expressed in muscle. Do the authors have any external evidence for these findings? On a different note, it became apparent that the authors did not report a molecular protein marker in the WB results.

We thank the review for highlighting this to us. Although previous studies in skeletal muscle cell cultures suggest that the cortical actin cytoskeleton is a major requirement for insulin-stimulated glucose transport, implicating the β-actin isoform, which in many cell types is the main actin isoform, nevertheless it is not clear that β-actin plays such a role in mature skeletal muscle. (https://www.ncbi.nlm.nih.gov/pmc/articles/PMC6087721/). Therefore, although it has been shown that b-actin should be excluded in studies including subjects with large age difference (doi: 10.1152/japplphysiol.00840.2014), our protein assays were using one cell line, therefore one donor, and this should not be a problem.

However, to address any concerns, we have revised the panels showing western blots and included the corresponding total protein (non-phosphorylated) for every western blot on figure 3.

We have added the size guide onto the WB images.

  1. The font of the labels on several y-axes is at minuscule levels. It is impossible to identify the numeric value of the y-variables.

We thank the reviewer for pointing this out, we have revised figure 3 accordingly.

Reviewer 2 Report

This study suggested that Sildenafil-induced CXCL10 inhibition at systemic and human muscle cell level supports and the PDE5inhibitor could be a potential therapeutic therapy to prevent and treat muscle damage in SSc. Results from this study are sound. A revision is suggested. 
1. Please discuss the clinical implications of this study.
2. For NF-kB study, the nuclear isolation is suggested. Please study the cytosolic and nuclear NF-kB and I-kB.
3. Please address the effects of sildenafil in pro-inflammatory diseases and address the mechanisms. 
4. Fig3. Please provide the blots of total protein , instead of actin. 
5. Please show the inhibitory effect of Sildenafil in PDE5 inhibition in this model.
6. The mechanism is still unclear. please should if si-CXCL10 affect the results.

Author Response

We thanks the reviewer for the positive comment on the soundness of our findings.

  1. Please discuss the clinical implications of this study.

Being exploratory of observational cohort we did not want to give too much clinical importance to our findings, which remain hypothesis generating. To address this comment  we added in the revised discussion:

We believe this is an hypothesis generating study, suggesting the potential scope of testing, in the context of a controlled trial, the use of sildenafil to mitigate muscle damage induced by SSc.

  1. For NF-kB study, the nuclear isolation is suggested. Please study the cytosolic and nuclear NF-kB and I-kB.

 We value the reviewers suggestion, however this is beyond the scope of this manuscript. And it would require to repeat the entire study. We acknowledge in the discussion:

Nuclear localisation of NF-kB and I-kB are warranted to confirm and extend our findings.

  1. Please address the effects of sildenafil in pro-inflammatory diseases and address the mechanisms. 

We value the reviewer’s suggestion, however there are not sufficient data to address this mechanisms and we would be unnecessarily speculative to do so. One of the aim of this study is exactly to inform basic and translational studies to address these mechanisms.

  1. Fig3. Please provide the blots of total protein, instead of actin. 

We value the reviewer’s comment, and have included the relevant total proteins (non-phosphorylated forms).

  1. Please show the inhibitory effect of Sildenafil in PDE5 inhibition in this model.

We thank the reviewer for this comment. The use of sildenafil in vitro has been proven effective to block PDE5 in multiple other studies and we did not perform any positive control in our experimental setting.

  1. The mechanism is still unclear. please should if si-CXCL10 affect the results.

We value the reviewers suggestion, while we do not have the capacity at the moment to include these experiments we acknowledge their importance in future studies in the discussion:

Future loss of function studies (siCXCL-10) looking at the autocrine effect of CXCL10 on myoblasts or myotubes may shed further light on the role of this molecule in muscle damage.

Round 2

Reviewer 1 Report

The last statement in the abstract section is still too strong for the study's limitations. It ought to be removed:

"Our data open a new frontier in the modulation of chemokine secretion and on the functional role of myocytes as cells directly involved in the immune response in SSc."

Also, the last statement in the discussion section needs better crafting with a mollifying factor included.

“In this sense, given its proven safety in patients with SSc, we believe that our study supports the rationale to explore in a controlled clinical experiment, the efficacy of sildenafil in aiding the management of inflammatory muscle involvement during SSc.”

Author Response

Many thanks for your comments. I have attached our responses.

Reviewer 2 Report

My questions had been well addressed, this submission is acceptable.

Author Response

Thank you for reviewing our responses.